# Reduction of Sulfoxides in Multigram Scale, an Alternative to the Use of Chlorinated Solvents

Laura Adarve-Cardona [ID] and Diego Gamba-Sánchez *[ID]

Laboratory of Organic Synthesis Bio- and Organocatalysis, Chemistry Department, Universidad de Los Andes, Cra. 1 No. 18A-12 Q:305, Bogotá 111711, Colombia; l.adarve@uniandes.edu.co
* Correspondence: da.gamba1361@uniandes.edu.co; Tel.: +57-60-1339-4949 (ext. 2691)

**Abstract:** In this manuscript, we describe the use of ethyl vinyl ether/oxalyl chloride as the reducing mixture for sulfoxides. The reaction is based on the high electrophilic character of chlorosulfonium salts, formed in situ by the reaction of oxalyl chloride and the sulfoxide. Thereafter, the nucleophilic vinyl ether acts as a chlorine scavenger, affording the corresponding sulfide. The method is applicable on a big scale and may be applied to highly functionalized sulfoxides. Chromatographic purification is only needed in exceptional cases of unstable substrates, and the final sulfide or the corresponding salt is usually obtained after simple evaporation of volatiles. The sole contaminants of this method are carbon dioxide, carbon monoxide and small (five-carbon maximum) aldol products, which are formed during the reaction process.

**Keywords:** sulfides; sulfoxides; sulfonium salts; reduction

## 1. Introduction

Academic laboratories constantly develop methodologies to perform organic reactions with better yields and conversions and cleaner reaction conditions. Most of these developments are widely applied in academic contexts, but sometimes there are issues with scaling them up and using them in industrial processes. Among other reasons, unsuitable solvents and chromatographic purification are some of the most critical issues. Notably, solvents are classified regarding their toxicity and the possibility that they may affect human health, and traditionally used solvents in academic laboratories are among the most toxic and unsuitable [1]. On the other hand, preparative chromatographic purifications are daily practices in academic contexts, but their use in big-scale transformations is expensive and produces significant amounts of waste. In that context, when academic researchers are confronted with the scale-up of organic reactions, they constantly face all the issues above, and additional investigation becomes necessary to improve existing methods and avoid the apparent disconnection between academic and industrial processes.

Due to its significance in pharmaceutical chemistry, the deoxygenation of sulfoxides has attracted the attention of many researchers [2–4], and striking green and scalable strategies have been reported to accomplish this fundamental transformation [5–9]. Unfortunately, a detailed and close analysis of some of these new alternatives always casts doubts on the possibility of scaling up those reactions in a pharmaceutical context. For instance, the recent work of Guo, Wen, and co-workers [5] describes an impressive electrochemical reduction in sulfoxides. The authors claimed the reaction might be performed on decagram scales. However, chromatographic purification is mandatory and the reaction solvent is class two, thus unsuitable in pharmaceutical industries.

We have recently faced these problems. Our recent work on the deoxygenation of sulfoxides is one of the most general and straightforward methods reported [10]. Unfortunately, it needs dichloromethane or chloroform as solvents and always produces a stoichiometric amount of chorotrimethoxybenzene, which should be separated by chromatography.

Inspired by these difficulties and after a close investigation of the reaction mechanism, we report an alternative, scalable method for reducing sulfoxides that produces no problematic residues, uses a class three solvent, and only requires chromatographic purification for unstable substrates.

## 2. Materials and Methods

### 2.1. Materials

All reactions were performed under an atmosphere of nitrogen. Acetone was dried over anhydrous $CaSO_4$ and distilled before use. All reagents were used as received from commercial suppliers. Reaction progress was monitored by thin-layer chromatography (TLC) performed on aluminum plates coated with silica gel F254, with 0.2 mm thick TLC plates visualized using ultraviolet (UV) light at 254 nm or stained with p-anisaldehyde, vanillin or $KMnO_4$ solutions. Flash column chromatography was performed using silica gel 60 (230–400 mesh). Omeprazole, lansoprazole, and albendazole were obtained as samples from commercial suppliers. Some compounds were purified using flash chromatography (FC). The solvents used for purification are described as follows: FC: (Solvent) or (Solvent 1 to Solvent 2) when a polarity gradient was applied.

### 2.2. Synthesis of Phenyl Propyl Sulfoxide (**7**)

Phenyl propyl sulfoxide was obtained following the previously reported procedure. All spectroscopic characteristics fit perfectly with previous reports [11].

[1]H NMR (400 MHz, $CDCl_3$) δ: 7.58–7.68 (m, 2H), 7.45–7.57 (m, 3H), 2.68–2.86 (m, 2H), 1.75–1.88 (m, 1H), 1.58–1.75 (m, 2H), 1.05 (t, *J* = 7.4 Hz, 3H).

[13]C NMR (100 MHz $CDCl_3$) δ: 144.2, 131.0, 129.3, 124.2, 59.4, 16.0, 13.4.

### 2.3. Reduction in Sulfoxides

To a solution of the corresponding sulfoxide (1 equiv.) and ethyl vinyl ether (1.5 equiv.) in dry acetone (0.15 M), oxalyl chloride (1.5 equiv.) was added slowly at room temperature and under a nitrogen atmosphere. The reaction mixture was stirred until complete conversion was observed by TLC (the total time of the reaction was around 30 min). Acetone and volatile impurities were removed under reduced pressure using a rotary evaporator at a vacuum pressure of 10 mbar and a water bath temperature of 45 °C. The virtually quantitative yield was determined by [1]H NMR of crude mixture. Gas chromatography analysis of the reaction mixture (including solvent) showed some short-chain organic impurities (see Supporting Information) undetected after evaporation. With some substrates, alternative purification by flash chromatography can be performed.

#### 2.3.1. Phenyl Propyl Sulfide (**8**)

According to the general procedure, phenyl propyl sulfoxide (1 mmol, 168.3 mg) was used to obtain 160 mg of light-brown oil as the reaction crude. When purified by flash chromatography using cyclohexane, the phenyl propyl sulfide (**8**) was obtained as a colorless oil (0.93 mmol, 141 mg, 93%).

[1]H NMR (400 MHz, $CDCl_3$) δ: 7.33 (d, *J* = 7.8 Hz, 2H), 7.27 (t, *J* = 7.7 Hz, 2H), 7.16 (t, *J* = 7.2 Hz, 1H), 2.90 (t, *J* = 7.3 Hz, 2H), 1.61–1.83 (m, 2H), 1.02 (t, *J* = 7.3 Hz, 3H).

[13]C NMR (100 MHz $CDCl_3$) δ: 137.0, 129.1, 128.9, 125.8, 35.7, 22.7, 13.6.

#### 2.3.2. Albendazole Hydrochloride (**12**)

According to the general procedure, ricobendazole (10.66 mmol, 3.0 g) was used to obtain 3.2 g of a light-yellow product (12.26 mmol, 3.2 g, 99%) after successive additions of solvent (acetone), re-evaporation, and drying in a Glass Oven Dryer (Vacuum laboratory drying oven/compact B-585 Büchi) at 100 °C and 10 mbar.

[1]H NMR (400 MHz, DMSO-$d_6$) δ: 7.50–7.59 (m, 2H), 7.32 (d, *J* = 10.2 Hz, 1H), 3.86 (s, 3H), 2.93 (t, *J* = 7.2 Hz, 2H), 1.49–1.63 (m, 2H), 0.96 (t, *J* = 7.3 Hz, 3H).

[13]C NMR (100 MHz DMSO-$d_6$) δ: 152.9, 144.5, 131.7, 130.7, 128.6, 125.4, 113.9, 113.4, 53.7, 35.4, 22.0, 13.2.

### 2.3.3. Reduction in Unstable Sulfoxides

To a solution of the corresponding sulfoxide (1 equiv.) and ethyl vinyl ether (1.5 equiv.) in dry acetone (0.15 M), oxalyl chloride (1.5 equiv.) was slowly added at room temperature and under a nitrogen atmosphere. The reaction mixture was stirred until complete conversion was observed by TLC (the total time of the reaction is around 30 min). Acetone was removed under reduced pressure, and the reaction mixture was neutralized with saturated aqueous NaHCO₃. The aqueous layer was extracted with AcOEt (3 × 20 mL). The combined organic extracts were washed with brine, dried over anhydrous Na₂SO₄, filtered, and concentrated under reduced pressure. The crude product may be purified using flash chromatography if impurities are observed.

### 2-(((3-Methyl-4-(2,2,2-trifluoroethoxy)pyridin-2-yl)methyl)thio)-1H-benzo[d]imidazole (**10**)

According to the general procedure, lansoprazole (**9**) (0.133 mmol, 49 mg) was used to obtain the desired product as a white solid without further purification (0.11 mmol, 40 mg, 85%).

[1]H NMR (400 MHz, CDCl₃) δ: 12.60 (s, 1H), 8.42 (d, *J* = 5.7 Hz, 1H), 7.54 (d, *J* = 65.8 Hz, 2H), 7.16–7.23 (m, 2H), 6.73 (d, *J* = 5.8 Hz, 1H), 4.42 (d, *J* = 8.3 Hz, 4H), 2.32 (s, 3H).

[13]C NMR (100 MHz CDCl₃) δ: 162.5, 157.7, 151.3, 147.5, 122.8 (q, $J_{C\text{-}F}$ = 277.9 Hz), 122.0, 121.8, 106.0, 65.5 (q, $J_{C\text{-}F}$ =36.5 Hz), 34.9, 10.7.

[19]F NMR (374 MHz CDCl₃) δ: −73.5 (s).

### 5-Methoxy-2-(((4-methoxy-3,5-dimethylpyridin-2-yl)methyl)thio)-1H-benzo[d]imidazole (**14**)

According to the general procedure, omeprazole (**13**) (5.5 mmol, 2 g) was used to obtain the desired product (**14**) as a brown oil. After purification by flash chromatography using pentane: AcOEt (9:1 to 6:4), a yellowish oil was obtained (5.13 mmol, 1.69 g, 89%).

[1]H NMR (400 MHz, CDCl₃) δ: 8.25 (s, 1H), 7.42 (d, *J* = 8.7 Hz, 1H), 7.04 (d, *J* = 2.4 Hz, 1H), 6.82 (dd, *J* = 8.7, 2.4 Hz, 1H), 4.35 (s, 2H), 3.85 (s, 3H), 3.78 (s, 3H), 2.32 (s, 3H), 2.28 (s, 3H).

[13]C NMR (100 MHz CDCl₃) δ: 165.4, 156.2, 155.9, 148.4, 126.7, 125.8, 111.4, 60.3, 56.0, 35.1, 13.6, 11.5.

### 2.4. Characterization Techniques

All [1]H NMR, [13]C, and [19]F NMR spectra were recorded using a BRUKER Avance III HD Ascend 400 spectrometer. Chemical shifts are given in parts per million (ppm, δ), referenced to the TMS ([1]H and [13]C) and trifluoroacetic acid ([19]F). When necessary, the solvent peak of residual CDCl₃ was defined at δ = 7.26 ppm ([1]H NMR) and δ = 77.16 ([13]C NMR). Coupling constants are quoted in Hz (*J*). Splitting patterns of [1]H NMR were designated as singlet (s), doublet (d), triplet (t), quartet (q), or multiplet (m). Splitting patterns that could not be interpreted or easily visualized were designated as multiplet (m) or broad (br). Copies of NMR spectra are provided as Supplementary Materials. GC-MS was recorded in a Thermo Scientific Trace 1300.

## 3. Results and Discussion

### 3.1. Optimization

As we mentioned in the introduction, the methodology described in this manuscript is based on our previous report on reducing sulfoxides. The summarized mechanism and the working hypothesis are depicted in Scheme 1.

**Scheme 1.** Reaction mechanism reported by Gamba-Sánchez et al. [10] and working hypothesis.

The reaction starts with the electrophilic activation of the sulfoxide **1**, yielding an unstable chlorosulfonium salt **3**, which immediately reacts with the nucleophilic trimethoxybenzene **4**, yielding the sulfide **5** and the chlorotrimethoxybenzene **6** after re-aromatization. This is the crucial step of our analysis and the improvement we propose herein. The most critical issue with our previous method was the production of **6**, since it needed to be removed by chromatographic techniques. We hypothesized that changing the trimethoxybenzene by a closely reactive species should allow us to carry out the same transformation (working hypothesis). However, the new nucleophile should have unique characteristics, be easy to handle, and have low toxicity.

Most importantly, the chlorinated product should be volatile or susceptible to being precipitated out of the reaction mixture at the end of the process. Having this in mind, we decided to start our studies using a solvent-free approach and several vinyl ethers or esters as nucleophilic reagents (entries 1 to 3, Table 1). The reaction proceeded in 30 min and at room temperature in all cases. However, a significant number of by-products were observed. All the nucleophiles reacted as enol equivalents, and auto-condensation and polymerization reactions were predictable. Nevertheless, the ethyl vinyl ether showed the cleanest crude under the described conditions.

**Table 1.** Optimization results using **7** and oxalyl chloride (1.5 equiv.).

| Entry | Nucleophile | Solvent | Concentration | Yield (%) | Observations |
|-------|-------------|---------|---------------|-----------|--------------|
| 1 | Vinyl acetate | N.A. | 0.15 M [1] | N.D. | Sulfide + unidentified mixture of polar products [2] |
| 2 | Ethyl vinyl ether | N.A. | 0.15 M [1] | N.D. | Sulfide + unidentified mixture of polar products [2] |
| 3 | Butyl vinyl ether | N.A. | 0.15 M [1] | N.D. | Sulfide + unidentified mixture of polar products [2] |
| 4 | Ethyl vinyl ether | *t*BuOMe | 0.15 M | N.D. | No conversion |
| 5 | Ethyl vinyl ether | DCM | 0.15 M | N.D. | Complete conversion, small impurities |
| 6 | Ethyl vinyl ether | Acetone | 0.15 M | N.D. | Complete conversion, clean product |
| 7 | Ethyl vinyl ether | Acetone | 0.50 M | N.D. | Complete conversion, medium impurities |
| 8 | Ethyl vinyl ether | Acetone | 1.00 M | N.D. | Complete conversion, dirty crude |
| 9 | Ethyl vinyl ether | Acetone | 0.15 M [3] | 93 | Product clean without further purification |

[1] The nucleophile acts as the solvent. [2] The primary product was the desired sulfide. We observed additional signals in the aliphatic zone of $^1$H NMR that probably corresponded to polymerization or autocondensation products from vinyl ether or ester. [3] The reaction was performed using 1 mmol of starting material.

We turned our attention to other solvents, starting with methyl tert-butyl ether (MTBE), considered one of the greenest solvents nowadays. Unfortunately, no reaction was observed (entry 4); even if MTBE has been designated as a proper solvent in many chemical

transformations, it has also been described to react with strong electrophiles, acting as a source of methyl and tertbutyl groups [12,13]. In that context, it is plausible to assume that the oxalyl chloride was consumed by reacting with the solvent, so no activation of sulfoxide proceeded. We used dichloromethane to have a reference point for comparing with our previous results. In this case, the product was formed close to quantitative yield, and few impurities were observed in the crude [1]H NMR (entry 5). Changing the solvent to acetone, a virtually pure product was observed (entry 6). We decided to concentrate the reaction mixture to reduce the solvent consummation; unfortunately, the impurities were more significant in those cases (entries 7 and 8). Finally, we used the conditions of entry 6 but used 1.0 mmol of the starting material. Once the reaction finished, the crude product was dried under reduced pressure, and pure sulfide was obtained with a 93% yield. It is worth mentioning that we used other substrates and solvents during our optimization studies. Observations of those let us conclude that there are no significant differences between solvents. However, the starting sulfoxide must be soluble to avoid practical issues and low conversions. With these results in hand, we decided to extrapolate the results to a polyfunctional sulfoxide of pharmaceutical interest and scale up the reaction.

### 3.2. Application to Substrates of Pharmaceutical Interest

We started studying the reduction in lansoprazole, a commercial drug used to treat gastroesophageal reflux, peptic ulcers, and Zollinger-Ellison syndrome. The reaction is shown in Scheme 2 and was performed using the reaction conditions optimized in Table 1.

**Scheme 2.** Reduction in lansoprazole.

The reaction proceeded smoothly and the product was obtained with excellent yield. There was no special treatment and the crude was evaporated to dryness. It is noteworthy that, since the reagent and the product have basic functions, they trap the HCl formed during the reaction. The product is isolated in its hydrochloride form, so the crude product was washed with a saturated solution of $Na_2CO_3$ to obtain neutral, reduced lansoprazole 10.

We then turned our attention to ricobendazole and omeprazole as substrates. Ricobendazole is an anti-parasite used in veterinary applications and its reduced form, albendazole, is used with similar purposes in humans. Therefore, the reduction was performed on different scales. First, we used 25 mg to obtain the free product (the neutral form). We observed that the action of $Na_2CO_3$ or $Et_3N$ might neutralize the hydrochloride; however, liquid-liquid extractions became mandatory and a slight loss of product was observed. The reaction performed at 1 mmol scale has similar behavior, but pure hydrochloride is obtained in quantitative yield if the crude reaction mixture is evaporated under reduced pressure. We observed an unidentified impurity during the treatment, removed by successive additions of solvent (acetone) and re-evaporation.

On the other hand, the reaction was performed at a multi-gram scale (3 g), obtaining the desired sulfide in quantitative yield. It is noteworthy that evaporation of the solvent afforded the sulfide with the impurity mentioned before, so we heated the hydrochloride at 100 °C under a vacuum for a couple of hours, obtaining the pure product without further purification. See Scheme 3.

**Scheme 3.** Reduction in ricobendazole.

Finally, we used commercially available omeprazole on a 2 g scale. The product aspect was bizarre; however, its spectroscopic data perfectly fit previous reports [14]. Nonetheless, we decided to perform a chromatographic purification to ensure the robustness of our method, providing the scientific community with several purification options depending on their needs. After purification, the free sulfide (neutral form) was obtained in 89% yield. Scheme 4.

**Scheme 4.** Reduction in omeprazole.

It is noteworthy that chromatographic analysis of the crude reaction mixture (see Supplementary Material) did not show chlorinated products, suggesting that after the reaction as the nucleophile, the chlorinated aldehyde product fragments into small, undetectable products.

## 4. Conclusions

We report herein a complementary scalable method to reduce polyfunctional sulfoxides. Compared with previously described methods, our alternative does not use metal catalysis, high-pressure gaseous material, or expensive and difficult-to-handle reagents. The reaction is performed using oxalyl chloride at room temperature, producing CO and $CO_2$ during the reaction, and ethyl vinyl ether produces volatile small organic molecules, so no pollutant by-products are formed. Consequently, this is one of the easiest methods for the reduction in sulfoxides described to date. It is applicable to sulfoxides of pharmaceutical interest, solids and liquids, and requires no special or chromatographic purification. When basic functions are present, the final product is obtained as a hydrochloride salt, but the neutral free molecule may be obtained by simple washing with $Na_2CO_3$ solution. The reaction uses a green and safe solvent and may be performed on multi-gram scale.

**Supplementary Materials:** The following supporting information can be downloaded at: https://www.mdpi.com/article/10.3390/pr10061115/s1, [1]H NMR spectra of all sulfides reported herein. GC-MS of the crude reaction mixture.

**Author Contributions:** Conceptualization, D.G.-S.; methodology, L.A.-C.; formal analysis, D.G.-S. and L.A.-C.; investigation, L.A.-C.; data curation, L.A.-C.; writing—original draft preparation, D.G.-S.; writing—review and editing, D.G.-S. and L.A.-C.; supervision, D.G.-S.; project administration, D.G.-S.; funding acquisition, D.G.-S. All authors have read and agreed to the published version of the manuscript.

**Funding:** This research was funded by Universidad de Los Andes, Faculty of Science, grant number INV-2021-128-2261.

**Institutional Review Board Statement:** Not applicable.

**Informed Consent Statement:** Not applicable.

**Data Availability Statement:** The data presented in this study are available upon request to the corresponding author.

**Acknowledgments:** L.A.-C. acknowledges the Universidad de Los Andes and especially the Chemistry Department for her fellowships. Andrés Cañon Ortiz is acknowledged for isolating some starting materials and preliminary experiments.

**Conflicts of Interest:** The authors declare no conflict of interest.

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
