# Peer review of "Reduction of Sulfoxides in Multigram Scale, an Alternative to the Use of Chlorinated Solvents"

_processes, doi:10.3390/pr10061115_

Round 1
Reviewer 1 Report
In recent paper the authors present their new results, which strongly connect to their previous paper (10.1002/chem.202001815), it is a kind of extension and basically, they only changed the applied solvents. The authors claim that they have a green process and clearly they replaced the chlorinated solvents to some others, but the application of a chlorinated compound (COCl)2 in one equivalent would not enable the term of "green". Therefore I strongly recommend changing the title and the text accordingly. Regarding the reaction mechanism: What could be the reason of the strongly dependence on the solvents? e.g. in case of tert-butyl methyl ether no reaction was detected. It should be discussed, it may increase the level of the paper.
Some minor suggestions: Scheme 1: The (COCl2) is not correct it should be (COCl)2. On top of that, "the oxidized function" and "oxidized intermediate" have different letter style than the other text on this figure.
Author Response
Reviewer 1
Q: In recent paper the authors present their new results, which strongly connect to their previous paper (10.1002/chem.202001815), it is a kind of extension and basically, they only changed the applied solvents. The authors claim that they have a green process and clearly they replaced the chlorinated solvents to some others, but the application of a chlorinated compound (COCl)2 in one equivalent would not enable the term of "green". Therefore I strongly recommend changing the title and the text accordingly.
Answer: According to the proposed reaction mechanism, the by-products generated by oxalyl chloride are CO and CO2, so no particular issues on sustainability must be mentioned. In other words, we believe that oxalyl chloride is a reagent that fits the green chemistry principles. However, we agree with the reviewer in terms of toxicity. Consequently, we changed the title and text according to the suggestion. Title and line 57
Q: Regarding the reaction mechanism: What could be the reason of the strongly dependence on the solvents? e.g. in case of tert-butyl methyl ether no reaction was detected. It should be discussed, it may increase the level of the paper.
Answer: We included a discussion about the solvents. It is worth mentioning that the effect of the solvent in the reaction is related to the solubility of the substrate. When the reagent (sulfoxide) is soluble, the differences between solvents are negligible. Concerning methyl tert-butyl ether, some reports in the literature show it as a reagent instead of a solvent, mainly reacting with strong electrophiles or acids, so we believe that oxalyl chloride reacts with the solvent forming volatile species and avoiding the reaction with the sulfoxide. References 12 and 13 were added to support our hypothesis. Lines 176 to 192
Q: Some minor suggestions: Scheme 1: The (COCl2) is not correct it should be (COCl)2. On top of that, "the oxidized function" and "oxidized intermediate" have different letter style than the other text on this figure.
Answer: Scheme 1 was entirely edited to follow this suggestion, the concern of the editorial office and the request of other reviewers.

Reviewer 2 Report
This paper submitted by Adarve-Cardona and Gamba-Sanchez described the reduction of sulfoxide using ethyl vinyl ether and oxalyl chloride as the reducing mixture. This method featured easy separation of the product and good tolerance on highly functionalized substrates. Although the reducing agent oxalyl chloride is very toxic, I still view this method as a scalable reduction reaction. However, there are still many issues in the paper that need to be addressed before acceptance.
1) In the first paragraph, the authors described how the internet is evolving and how it affects chemistry development. I do not think this is relevant and necessary.
2) I suggest the author consider adding a scheme to describe their previous work (ref. 14)
3) In Line 79, when the authors describe the reaction, they should specifically point out which reagent is the limiting reagent and add the volume of acetone.
4) I am confused about how the authors reference their NMR spectra, TMS, or solvent residue?
5) I suggest the author polish scheme 1, the arrows and some bonds are distorted.
6) Table 1, authors should add more detail to the chemical equation.
7)scheme 2, the amount of the substrate should be added to the equation.
Author Response
Reviewer 2
Q: In the first paragraph, the authors described how the internet is evolving and how it affects chemistry development. I do not think this is relevant and necessary.
Answer: The first paragraph of the introduction was changed.
Q: I suggest the author consider adding a scheme to describe their previous work (ref. 14)
Answer: We appreciate the reviewer’s suggestion; however, we think an additional scheme will be redundant. Alternatively, we edited scheme 1, indicating the previous and current work.
Q: In Line 79, when the authors describe the reaction, they should specifically point out which reagent is the limiting reagent and add the volume of acetone.
Answer: The experimental section describes a general procedure, so the quantity of solvent depends on the amount of the limiting reagent (1.0 equivalent). The procedure specifies that sulfoxides are the limiting reagent since we always used 1.0 equivalents. We also used the equivalents of other reagents (oxalyl chloride and vinyl ether) and pointed out the concentration of the limiting reagent in acetone. So, the demanded information is already included. On the other hand, there is no notice in information for authors (or author guidelines from the journal) that forbid writing a general procedure, so we cannot attend to the reviewer’s suggestion.
Q: I am confused about how the authors reference their NMR spectra, TMS, or solvent residue?
Answer: NMR spectra are always referenced to TMS, which has an assigned shift of 0 ppm. When the TMS signal is too small, or people use solvents without TMS, they use the residual solvent to reference the spectra. Nevertheless, the value is relative to TMS. In other words, the reference is TMS even if you don’t have its signal in the NMR. We changed the sentence in the experimental section to avoid confusion: “Chemical shifts are given in parts per million (ppm, δ), referenced to the TMS (1H and 13C) and trifluoroacetic acid (19F), when necessary, the solvent peak of residual CDCl3 was defined at δ = 7.26 ppm (1H NMR) and δ = 77.16 (13C NMR) ”
Q: I suggest the author polish scheme 1, the arrows and some bonds are distorted.
Answer: Scheme 1 was entirely edited. The schemes were designed using ChemDraw software and the manuscript using IOS. Sometimes, there are compatibility issues that distort the schemes or bonds. So, we hope that this issue will be arranged with a .pdf version of the manuscript and in the final editing process. However, we carefully checked the schemes and will provide original files in the submission process.
Q: Table 1, authors should add more detail to the chemical equation.
Answer: The equation of table 1 was edited, and the missing information added
Q: Scheme 2, the amount of the substrate should be added to the equation.
Answer: Schemes 2, 3 and 4 have been edited, and the quantity of starting material added, the schemes titles were changed accordingly.

Reviewer 3 Report
Introduction is not written professionally
10 first lines are not in the subject:
With the introduction and massification of the Internet back in the 1990s,[1] the pro- 20 cess for scientific publication suffered a significant change; quick electronic submissions, access to worldwide scientific peers, track the whole publication processes, among others, are now common in scientific vocabulary.[2] Also, the average time from submission to publication was significantly reduced in all areas of knowledge.[3] Thanks to these developments, scientists have lots of new information daily, and logically chemistry is not alien to this event. Based on those improvements, one of the most significant advances in organic chemistry is the daily report of new methodologies aiming to access complex organic functions and make reactions more selective. Unfortunately, the world of academic developments and the pharmaceutical industry is somehow disconnected.[4] Despite the marvellous results in enantioselective transformations, chemoselective reactions, newtechniques such as alternative sources of energy, photocatalysis and many others, the transition between the academic laboratory and the industrial process is generally problematic.
Also the references are not in the aim of this research.
- Naughton, J. The evolution of the Internet: from military experiment to General Purpose Technology. J. Cyber Policy., 2016, 1, 5- 252 28. DOI: 10.1080/23738871.2016.1157619 253
- Yin, D.; Tam, W.L.; Ding, M.; Tang, J. MRT: Tracing the Evolution of Scientific Publications. IEEE Trans. Knowl. Data. Eng., 2021, 254 1-1. DOI: 10.1109/TKDE.2021.3088139 255
- Powell, K. Does it take too long to publish research? Nature, 2016, 530, 148-151. 10.1038/530148a 256
- Michaudel, Q.; Ishihara, Y.; Baran, P.S. Academia–Industry Symbiosis in Organic Chemistry. Acc. Chem. Res., 2015, 48, 712- 721. 257 DOI: 10.1021/ar500424a 258
- Witschi, C.; Doelker, E. Residual solvents in pharmaceutical products: acceptable limits, influences on physicochemical properties, analytical methods and documented values. Eur. J. Pharm. Biopharm., 1997, 43, 215-242. DOI: 10.1016/S0939-6411(96)00037- 260 9
The
Abstract is not accurate and concise. The approach/ methods are not properly described.
The conclusions and interpretations are not sound. I really don’t know whether it is within the scope of the journal. Anyway, although Illustrations or Drawings is adequate and subject is novel, but "Technical Quality" is not sufficient. Style & Overall Representation is not good at all. The previous study is not extended at all and gap of research has not been described.
Suitability for Journal could be final decision of editorial board.
I also checked similarity with ithenticate software which results showed 23% overlap.

Author Response
Reviewer 3
Q: Introduction is not written professionally
10 first lines are not in the subject:
With the introduction and massification of the Internet back in the 1990s,[1] the process for scientific publication suffered a significant change; quick electronic submissions, access to worldwide scientific peers, track the whole publication processes, among others, are now common in scientific vocabulary.[2] Also, the average time from submission to publication was significantly reduced in all areas of knowledge.[3] Thanks to these developments, scientists have lots of new information daily, and logically chemistry is not alien to this event. Based on those improvements, one of the most significant advances in organic chemistry is the daily report of new methodologies aiming to access complex organic functions and make reactions more selective. Unfortunately, the world of academic developments and the pharmaceutical industry is somehow disconnected.[4] Despite the marvellous results in enantioselective transformations, chemoselective reactions, newtechniques such as alternative sources of energy, photocatalysis and many others, the transition between the academic laboratory and the industrial process is generally problematic.
Answer: We disagree with the reviewer. First, “professionally” is a subjective and relative term highly associated with the writer’s background. We wanted to contextualize the reader in a common problem for organic chemists. However, since reviewer 1 also expressed his/her reluctance to the introduction, we changed the first lines and aborded the subject more traditionally.
Q: Also the references are not in the aim of this research.
Naughton, J. The evolution of the Internet: from military experiment to General Purpose Technology. J. Cyber Policy., 2016, 1, 5- 252 28. DOI: 10.1080/23738871.2016.1157619 253
Yin, D.; Tam, W.L.; Ding, M.; Tang, J. MRT: Tracing the Evolution of Scientific Publications. IEEE Trans. Knowl. Data. Eng., 2021, 254 1-1. DOI: 10.1109/TKDE.2021.3088139 255
Powell, K. Does it take too long to publish research? Nature, 2016, 530, 148-151. 10.1038/530148a 256
Michaudel, Q.; Ishihara, Y.; Baran, P.S. Academia–Industry Symbiosis in Organic Chemistry. Acc. Chem. Res., 2015, 48, 712- 721. 257 DOI: 10.1021/ar500424a 258
Witschi, C.; Doelker, E. Residual solvents in pharmaceutical products: acceptable limits, influences on physicochemical properties, analytical methods and documented values. Eur. J. Pharm. Biopharm., 1997, 43, 215-242. DOI: 10.1016/S0939-6411(96)00037- 260 9
Answer: The first references were erased since the introduction was changed. However, the last one is still in the text because it is related to the classification of organic solvents and the acceptance of their residues in pharmaceutical products.
Q: The Abstract is not accurate and concise.
Answer: The abstract was edited, and we avoided contextualization, trying to make it more concise.
Q: The approach/ methods are not properly described.
Answer: The methods are described as usual in organic chemistry, all details are provided, and technical results are expressed as usual. We don’t understand the reviewer’s concern.
Q: The conclusions and interpretations are not sound.
Answer: We edited the conclusions traying to be more specific and highlight our results since it was demanded by reviewer 4. However, we strongly disagree in terms of interpretation of the results. Evidence is provided as usual in academic papers. However, if the reviewer has academic or scientific arguments, instead of merely personal opinions, we will be happy to follow the recommendations or discuss them properly.
Q: I really don’t know whether it is within the scope of the journal. Anyway, although Illustrations or Drawings is adequate and subject is novel, but "Technical Quality" is not sufficient. Style & Overall Representation is not good at all. The previous study is not extended at all and gap of research has not been described.
Suitability for Journal could be final decision of editorial board.
Answer: The suitability of the journal is, indeed, a decision of the Editorial office. Besides, this paper was submitted to a special issue that aims to connect academic and industrial practices. If the designed editors find a manuscript unsuitable for publication in this journal, they usually do not send it for peer review. Concerning the technical quality, style and representation, etc., there are no academic or scientific arguments in the particular case of these comments, only personal opinions that should be avoided in peer evaluation. The role of a reviewer is to express his/her academic concerns and help improve the manuscript and the evolution of science; it is not purely expressing his/her opinion without arguments. We did not make any change in the manuscript oriented by these comments.
Q: I also checked similarity with ithenticate software which results showed 23% overlap.
Answer: This is a rude and unnecessary accusation; it is made without proper arguments. The manuscript is original and is not published or under consideration elsewhere. I would like to call the reviewer's attention to the use of text analysis software. They check for similarity and not for textual reproduction. After reading this comment, we made the exercise ourselves; if the manuscript is copied and pasted in that software, a similarity of around 20% is observed. However, if you do the same exercise without using references and the experimental section, the similarity decreases to nearly 10%. Checking the details of the analysis, it is evident that the software overlaps commonly used phrases.
In summary, if the reviewer has any concerns about plagiarism, he/she should strongly recommend the editor in chief follow proper actions to prevent the publication of this paper. Otherwise, these comments do not help the publication process or improve the manuscript. I respectfully ask the editorial office to check the manuscript with the proper tools. If there is any doubt about the originality of this manuscript or the reported data, please let me know to discuss it correctly and without aggression or accusations. Again, we did not change the manuscript following this comment.

Reviewer 4 Report
This paper is well-written, with data that is easily understood by the reader; nonetheless, minor modifications are recommended before publishing.Lines 20-26 - The writing is too general and redundant.
I have noticed subscripts are missing in many chemicals’ formulas (ex: Lines 62-66)
Scheme 1 – Please include the electron lone pairs on -OMe oxygens
Table 1 - By products names has to be included
Line 197 – replace “afford” with “obtain”
Conclusion is too brief and weak - please improve it
Author Response
Q: Lines 20-26 - The writing is too general and redundant.
Answer: The introduction was changed according to the other reviewer’s suggestion. In consequence, lines 20-26 were removed. Hopefully, the new version of the introduction will be less general and less redundant.
Q: I have noticed subscripts are missing in many chemicals’ formulas (ex: Lines 62-66)
Answer: We corrected the formulas as demanded.
Q: Scheme 1 – Please include the electron lone pairs on -OMe oxygens
Answer: Scheme 1 was entirely edited, and we included lone pairs or OMe and in the vinyl ether to enhance the comprehension of similarity between functions.
Q: Table 1 - By-products names has to be included
Answer: We did not characterize the by-products of these reactions, so we cannot provide names or structures. However, we observed signals in the aliphatic zone of 1H NMR spectra, probably due to polymerization or auto condensation of vinyl ethers or esters. We avoided the word degradation, which is common in these cases, because the starting material reacted, and the products are from secondary reactions of the solvent. To avoid misunderstandings, we changed the word by-products to “unidentified mixture of polar products” and added a note with an explanation to the table.
Q: Line 197 – replace “afford” with “obtain”
Answer: Changed
Q: Conclusion is too brief and weak - please improve it
Answer: The conclusion was edited and improved; we hope the new version is more robust and specific.

Round 2
Reviewer 1 Report
The paper improved significantly.
Reviewer 2 Report
In this revised version, the authors addressed most of my and other reviewers' concerns. I think this article meet the critiria of this journal.
Reviewer 3 Report
I listed the flaws, and I don't believe that the manuscript has substantially improved in this short time.